# Traditional Bullying and Discriminatory Bullying Around Special Educational Needs: Psychometric Properties of Two Instruments to Measure It

**DOI:** 10.3390/ijerph16010142

**Published:** 2019-01-07

**Authors:** Antonio J. Rodríguez-Hidalgo, Anabel Alcívar, Mauricio Herrera-López

**Affiliations:** 1Department of Psychology, Universidad de Córdoba, 14071 Córdoba, Spain; 2Department of Education, University Laica Eloy Alfaro of Manabí, Manta 130802, Ecuador; gloria.alcivar@uleam.edu.ec; 3Department of Psychology, University of Nariño, San Juan de Pasto 52001, Colombia; mherrera@udenar.edu.co

**Keywords:** bullying, discrimination, aggression, victimization, special educational needs, disabilities

## Abstract

Two important challenges in research on bullying are to have reliable tools to measure traditional bullying and discriminatory bullying related to special educational needs (SEN), and to learn more about their prevalence. We present the validations of two instruments to measure bullying (European Bullying Intervention Project Questionnaire, EBIPQ) and discriminatory bullying with respect to SEN (EBIPQ–Special Education Needs Discrimination version, henceforth EBIPQ-SEND). A total of 17,309 teenagers from Ecuador took part in the study (M = 14.76, SD = 1.65; 49.9% male). The item response theory analyses evidenced accuracy and quality of the measures. The confirmatory factor analyses of the EBIPQ and the EBIPQ-SEND revealed the same two-factor structure—aggression and victimization—regardless of gender, showing optimal fit indexes. We present the results of the prevalence according to the roles of participation in traditional bullying and discriminatory bullying around SEN. Significant gender and age differences were observed for involvement in both phenomena. We also discuss the advantages of applying the EBIPQ and the EBIPQ-SEND to evaluate and diagnose harassment and discriminatory harassment around SEN.

## 1. Introduction

Bullying negatively conditions the development and health of the minors involved, as well as the processes of school teaching/learning and the institutional environment [1]. It has been observed that students with special educational needs—henceforth SEN—present a special risk of involvement in bullying and discrimination [2,3]. Regarding discriminatory bullying around SEN or disabilities, there is a scarcity of studies at a global level. It is necessary to have measurement instruments with optimal psychometric and internationally equivalent properties to measure traditional bullying and discriminatory bullying around SEN, as they will help us to advance the understanding of these phenomena in different countries.

### 1.1. Traditional Bullying: Characteristics, Measurement, and Prevalence

The term bullying has changed dramatically over several centuries, with its meaning defined based on social and historical contexts, and with the contribution of researchers, and community members, both adults and children [4]. Bullying is the systematic abuse of power among peers that causes physical, emotional, social and/or educational harm [5]. Its origin lies on the deliberate and repeated aggressions occurring under a power imbalance between the bully and the victim [6]. Bullying presents two dimensions: aggression and victimization [7], where the following roles of involvement appear: bully, victim, bully/victim, and non-involved (e.g., [8]). In the last decade, labels such as “traditional bullying” have been used to refer to its general and non-specific nature [9]. The developmental dynamics of aggression involves individual and environmental factors [10].

In the last years, the European Bullying Intervention Project Questionnaire (EBIPQ) [11] has allowed researchers to measure the two dimensions of bullying by means of two subscales: victimization and aggression. This instrument has been validated and has evidenced very good psychometric properties in European countries such as Spain [8,12]. Furthermore, certain validations of this instrument are being developed in Latin American countries such as Colombia or Panama [13].

In the last decade, the number of studies on bullying in Latin America has increased. In the case of Nicaragua, it has been observed that the prevalence of bullying in the last years of primary school was estimated at 50%, distributed by roles as follows: 25.3% victims, 6% bullies, and 18.7% bully/victims [14]. In this last study, boys stood out as bullies and bully/victims in opposition to girls. Nevertheless, the prevalence of bullying in secondary school in Nicaragua was estimated at 35%, distributed by roles as follows: 12.4% victims, 10.9% bullies, and 11.7% bully/victims [15]. In this last case, girls stood out as victims and bullies in comparison to boys, who were more likely to be bully/victims. In a study carried out in Colombia, an adaptation of the EBIPQ instrument was used upon a sample of secondary school adolescents, reporting a prevalence of bullying of 41.9%, distributed by roles as follows: 23.4% victims, 4.5% bullies, and 14% bully/victims [13]. In all these roles, boys were more involved than girls.

A transnational study in which 33 European and North American countries took part [16] showed that the average involvement as a victim of bullying (victims and bully/victims) reached 11.3%, quite below the evidenced average in the studied countries of the region of Latin America. A study that was carried out in Spain using the EBIPQ reveals that 18.8% are victims, exceeding the European average (19.4% are victims and 12.5% are bully/victims), and that 6.3% are pure bullies [8].

### 1.2. Bullying and Special Educational Needs (SEN): Singularities and Measurement

The scientific literature describes that the students with SEN are more involved in traditional bullying than the rest of their peers without SEN [17,18]. The students with SEN stand out in the roles of victim [19], bully [20], and in the double role of bully/victim [21]. In the intimidation process, it was observed that students with SEN can take on different roles which may vary [22,23]. The involvement of minors with SEN in bullying is associated with certain difficulties in the academic adjustment [23,24] and the psychosocial adjustment [25].

The prevalence of traditional bullying—aggression and/or victimization—of children and adolescents without SEN is between the 30% and 50% [5,17,26]. Nevertheless, students with SEN show a higher prevalence of bullying (between 1 and 1½ times higher) than their peers, which vary following their different cognitive–behavioral, physical, sensory, social, and/or communicative profiles [18,27].

Three research lines are followed on the issue of bullying and SEN, each one characterized by the use of a type of measurement instrument: (1) studies on bullying in a general minor population who are distinguished in terms of having SEN or not [3,25,26,28,29]; (2) specific studies on bullying in minors with SEN [23,27,30]; and (3) studies of diagnosis and clinic observation in cases of SEN that consider bullying experiences (e.g., [31]).

Most studies on bullying based on a general population that discriminate in terms of having SEN or not have used self-report questionnaires derivative from the Olweus Bully/Victim Questionnaire [6] and have focused on the victimization for bullying (e.g., [26]). Some research has used other instruments, using several items or even Likert-type scales of aggression and/or victimization among peers for the report of traditional bullying (e.g., [3,25,29]).

The specific studies on the bullying in which students participate commonly use new hetero-report or self-report questionnaires as well as adaptations of existing instruments (e.g., [23,30]). For example, the Bullying Behavior and Experience Scale (BBS) [27] is used to measure aggression and victimization in students with SEN regarding their peers in terms of verbal (direct and indirect) violence.

Other studies on bullying regarding SEN students focus on the development of instruments for the functional diagnosis of some types of specific SEN, such as autism (e.g., [31]). Some of these instruments implement certain items that allow us to identify bullying experiences (e.g., [32]). The three research lines described regarding SEN have focused their attention fundamentally on the study of traditional bullying. This has allowed us to learn that SEN is a predictor of direct involvement in bullying [17,22,33]. However, while knowing that it is quite common to find discrimination actions against students with SEN among peers [21,34], there are not many studies that have globally focused on the study of discriminatory bullying towards the individual due to SEN.

### 1.3. The Present Study

The study of bullying in Latin American countries on wide samples by means of reliable and validated measurement instruments, equivalent to the ones used in other countries from other regions, is a scientific challenge. Furthermore, another challenge is to identify and learn more about the particularities of the aggression and victimization in school-aged children with SEN in order to prevent and reduce it [28,35]. This second challenge is conditioned by: (1) the absence of validated and standardized instruments that assess bullying in students with SEN [27]; and (2) the scarcity of studies on discriminatory bullying towards individuals with SEN [23]. Whether we had reliable instruments, we could also know the extent of this phenomenon in the school population.

Therefore, the present study has the purpose of overcoming these challenges. The first objective is to determine the psychometric properties of the EBIPQ and to assess its validate as a measure of traditional bullying in a sample composed by Ecuadorian adolescents. The second objective is to validate the adaptation of the EBIPQ in Ecuador to measure discriminatory bullying around SEN. The third objective is to learn more about the prevalence of traditional bullying and discriminatory bullying around SEN in Ecuadorian adolescents regarding their roles of victim, bully, and bully/victim, as well as their possible gender variations. The first hypothesis of the study is that the EBIPQ will show optimal psychometric properties for a two-factor structure identical to the original version. The second hypothesis is that the adaptation of the EBIPQ to measure discriminatory bullying around SEN will show optimal psychometric properties for a two-factor structure identical to the one in the original instrument.

## 2. Materials and Methods

### 2.1. Participants

The study population were students from the eighth, ninth and tenth grades of the Ecuadorian Basic Education and from the first, second, and third grades of the Ecuadorian Baccalaureate in the 4th area of Ecuador, western region between Manabí and Santo Domingo de los Tsáchilas. We selected a representative sample of this population depending on the proportion of the population in the six levels and their affiliation to public, semi-private and private schools. The sample was composed of 17,309 students from 44 schools. The students were 49.9% were male and 50.1% were female, with ages ranging from 11 to 20 years (M = 14.76; SD = 1.65). Specifically, 12.8% of the students were in the eighth grade, 14.2% were in the ninth grade, and 17.1% in the tenth grade of Basic Education. On the other hand, 19.1% were in the first grade of Baccalaureate, 19.8% were in the second grade, and 17% were in the third.

### 2.2. Instruments

The self-report questionnaire has a section that collects sociodemographic information related to age, sex, characterization of degree and educational center; it is also requested, the self-identification of having SEN and marking the type of SEN or disabilities (e.g., visual impairment, hearing impairment, autism, learning difficulties, attention deficit hyperactivity disorder, emotional and behavioral difficulties).

To measure bullying, the European Bullying Intervention Project Questionnaire (EBIPQ) [11] was used, translated from English into Spanish [12]. It starts with an introductory question that guides the sense of response to the items: “You will be asked about the possible experiences related to bullying in your environment (school, friends and acquaintances). Have you ever experienced any of the following situations in the last two months?” Then, we asked them to respond to each one of the 14 items—7 for victimization and 7 for aggression, equivalent between them—by means of a Likert-type scale with five response options from 1 to 5 (these being 1 = never, 2 = once or twice, 3 = once or twice a month, 4 = around once a week, 5 = more than once a week). An example of victimization item is: “Someone has hit me, kicked me or pushed me”. An example of aggression item is: “I have hit, kicked or pushed someone”. The values of internal consistency of the original test were acceptable and revealed a high degree of test-retest reliability: αT1victimization = 0.84, αT2victimization = 0.88, αT1bully = 0.73, and αT2bully = 0.69 [11].

To measure discriminatory bullying towards SEN, an adaptation of the EBIPQ was used [11,12]. The adaptation consisted of modifying the explanation and the introductory question that guide the sense of response to the items: “Now you will be asked about your possible experiences of discrimination in your environment (school, friends, acquaintances) due to disabilities or SEN differences. Have you ever experienced any of the following situations in the last two months?” We kept the same wording of the 14 items—7 of victimization and 7 of aggression—and the same way of response from the original scale, as they represent a thorough range of behaviors of victimization and aggression that can be experimented or developed in a discriminatory way.

### 2.3. Procedure

The research had a cross-sectional, retrospective, ex post facto design, with one group and multiple measures [36] that respect the ethical principles of the Declaration of Helsinki of the World Medical Association as well as the national laws that regulate the research and the psychologist’s profession. Procedure was approved by the ethic committee of the University of Córdoba (PSI2016-74871-R). Due to ethical considerations, we proceeded to obtain authorization from the Ministry of Education of Ecuador through the 4th Zonal Coordination. With the administration team the application of the questionnaires was coordinated and the informed consent duly signed by the families was obtained. Then, from the second half of the school year 2016–2017, we visited the schools to distribute the questionnaire. The students were explained the purpose of the study and we insisted on the anonymous, confidential and voluntary nature of their participation. Students with SEN or disabilities who needed help to respond received support from the interviewers, few students needed it. The average time for filling up the questionnaire was 30 min. To obtain the Ecuadorian adaptation, the instruments were firstly submitted to a content validation by means of the expert judgment. These experts assessed the criteria of: vocabulary adaptation, concept clarity, coherence, and relevance of each of the items. For doing so, we used a 4-point Likert-type scale (1 = it does not fulfill it, 2 = low level of fulfillment, 3 = moderate fulfillment, and 4 = high level of fulfillment). Finally, we carried out a pilot test with 356 students in order to evaluate the level of understanding of the items. It was not necessary to modify any of the items as the participants proved to understand them well.

In order to establish the different roles of involvement, we followed the criteria established by the authors of the scales [11,37]. This way, to determine the victim role, we considered those individuals who got equal or lower scores to 3 (once a month) in any of the items of victimization and who got equal or lower scores to 2 (once or twice) in all the items of aggression. The involvement in the role of bully was calculated taking into account those individuals who got equal or higher scores to 3 (once a month) in any of the items of aggression, and who got equal or lower scores to 2 (once or twice) in all the items of victimization. The role of victimized-bully was calculated considering those individuals who got equal or higher to 3 (once a month) in at least one of the items of aggression and of victimization.

### 2.4. Data Analysis

A Mardia’s coefficient analysis was carried out to determine the presence or the absence of multivariate normality of the dataset through the R program [38] using MVN 1.6 (version, company, city, country) [39].

To verify the psychometric properties of the EBIPQ, we initially carried out analyses from the item response theory (IRT), calculating a three-parameter model (3PL) fitted to polytomous scales [40], which offers values of discrimination (a), difficulty (b), and probability of success or failure (c), for each item.

For the construct validity, confirmatory factorial analyses (CFAs) were carried out using an EQS 6.2 program (company: Multivarite Software Inc., city, country) [41]. For these analyses, we chose the method of weighted least squares estimation with a robust scaling [42] and the use of polychoric correlations [43], recommended for variables of categorical nature and absence of multivariate normality. In order to assess the fit of the models, the following indexes were used: Satorra–Bentler scaled chi-squared (χ2S-B) [44], the comparative fit index (CFI), the non-normality fit index (NNFI) (≥0.90 is adequate; ≥0.95 is optimal), the root mean square error of approximation (RMSEA) (≤0.08), and the root mean square residual (SRMR) (≤0.08 is adequate; ≤0.05 is optimal) [45]. We also assessed the Akaike information criterion (AIC) to compare the proposed models, being better the one with a lower value.

In order to learn more about the generalization of the model—the degree of robustness of the factorial structure—a configuration invariance analysis was performed establishing gender as the analysis criterion. This analysis consists of the comparison between the fit indexes of the models following the gender and the general model. The configuration invariance was assessed taking into consideration the delta (Δ) values of the NNFI, CFI, RMSEA, and SRMR fit measures, adopting a change ≤0.01 as a cut-off point in order to accept the invariance hypothesis [46]. Finally, the chi-squared difference test (Δχ2S-B) was used, where non-significant differences show invariance in the models [47,48]. This multi-group analysis was carried out with an EQS 6.2 program [41].

The analysis of internal consistency was carried out with the McDonald’s Omega index (ω), recommended for categorical variables and with the absence of multivariate normality [49], calculated with the Factor 9.2 program (company, city, country) [50]. The composite reliability (CR) was also determined, which indicates the general reliability of the set of items. The considered cut-off value for the composite reliability was 0.70 [51].

To identify the significant differences between the prevalence of the roles depending on the gender and age, we carried out proportion contrast tests (χ2), bearing in mind the values of the corrected typified remainders that were higher than +1.96 (confidence interval -CI-: 95%) and +2.58 (confidence interval: 99%). We considered the phi value for contrasts between two proportions 2 × 2, and the contingency coefficient index to test 2 × 3. The adopted statistical significance level was 0.5.

## 3. Results

### Validation of the Test

Mardia’s analysis generated a coefficient of skewness of 98.672; *p* > 0.05 and a coefficient of kurtosis of 282.472; *p* > 0.05, indicating the failure to meet the assumption of multivariate normality of data.

The analysis based on the IRT showed discrimination values higher to 0.71 and 1.97, considered from moderate to very high levels; the difficulty degree of the items ranged from 0.08 to 1.93, considered as acceptable levels (from 4 to 4); and the values of the probability of failure or success were low, indicating a good quality of the items [52] (see Table 1). The polychoric inter-item correlations also showed optimal values that report the consistency and quality of the construct (see Table 2).

The total internal consistency and the consistency of each factor of the EBIPQ scale—victimization (VB) and aggression (AB)—were optimal (ωVB = 0.81, ωAB = 0.76, ωtotal-EBIPQ = 0.83). To perform the CFA, we initially hypothesized a one-factor structure obtaining non-fit indexes: χ2S-B = 2959.846; *p* < 0.01; NNFI = 0.928; CFI = 0.939; RMSEA = 0.047 (90% CI (0.045, 0.048)); SRMR = 0.098; AIC = 2805.84. Later, we tried the original structure of two-factors obtaining optimal fittings reflected in the improvement of all the indexes, especially in the RMSEA and AIC, in addition to adequate factor weights and measurement errors: χ2S-B = 1258.800; *p* < 0.01; NNFI = 0.950; CFI = 0.954; RMSEA = 0.041 (90% CI (0.039, 0.042)); SRMR = 0.080; AIC = 1106.800 (see Figure 1). These results show that the two-factor model is the most suitable for Ecuador.

The CFA model of the delimited EBIPQ to report discrimination around SEN also led to optimal fit indexes χ2S-B = 2098.196; *p* < 0.01; NNFI = 0.967; CFI = 0.972; RMSEA = 0.068 (90% CI (0.066, 0.069)); SRMR = 0.064; AIC = 1946.197 (see Figure 2). The total internal consistency as well as the consistency of each factor was optimal (ωVB-SEN = 0.89, ωAB-SEN = 0.92, ωEBIPQ-SEN-Total = 0.93).

The composite reliability (CR) indexes of the victimization (VB) and aggression (AB) dimensions, both for the original EBIPQ and for the adapted scale to report discrimination around SEN, showed optimal values: CREBIPQ-VB = 0.94; CREBIPQ-AB = 0.96; CREBIPQ-VB-SEN = 0.88; CREBIPQ-AB-SEN = 0.92 [53].

The results of the multi-group analysis indicated invariance of the factor structure with respect to gender (see Table 3).

The chi-square differences (χ2S-B) between the models compared in the multi-group analysis were not significant and the delta values (∆) of the CFI, NNFI, RMSEA, and SRMR indexes did not exceed the limit of ≤0.01 established to accept the factorial invariance in all the comparisons [46] (see Table 3).

The calculation of prevalence showed that 19.6% are involved as victims, 6.3% as bullies and 20.4% as bully/victims regarding bullying. The total prevalence of involvement in bullying was 46.3%. Regarding gender, the test of proportion contrast showed significant associations in all the roles of involvement. Girls were shown to be more victims (χ2 (1, 17,293) = 9.817; *p* < 0.05 (phi = 0.024; *p* < 0.05)), whereas boys turned out to be more bullies (χ2 (1, 17,295) = 24.944; *p* < 0.001 (phi = −0.038; *p* < 0.001)) and bully/victims (χ2 (1, 17,300) = 121.860; *p* < 0.001 (phi = −0.084; *p* < 0.001)).

The 30.4% of the total of the sample admitted being involved in the discriminatory bullying around SEN. The involvement in the role of victimization did not show significant differences, which indicates that both men and women are undifferentiated victims of bullying. Regarding the aggression, a greater involvement on the part of the males is recognized (χ2 (1, 850) = 7.680; *p* <0.01 (phi = −0.021; *p* < 0.01)), as in the role of victimized aggressor (χ2 (1, 2496) = 58.114; *p* < 0.001 (phi = −0.058; *p* < −0.001)). Intermediate adolescence (14 to 16 years) refers to the stage of evolutionary development with greater involvement in the roles of victimization (χ2 (9, 1929) = 23.933; *p* < 0.05 (contingency coefficient = 0.037; *p* < 0.05)) and aggressor-victimized (χ2 (9, 2497) = 74.502; *p* < 0.001 (contingency coefficient = 0.066; *p* < 0.001)) (see Table 4).

## 4. Discussion

Our first objective was to check the validity of the EBIPQ [11,12] for Ecuador. The item response theory (IRT) analyses evidence that the items of the scale are sensible to the register of different behaviors, which are well-understood and which can be answered in an appropriate way by adolescents in Ecuador. The confirmatory factorial analysis (CFA) evidences the two-factor structure—victimization and aggression—in the measurement of traditional bullying. We can highlight the quality of the measurement of each one of the items and its contribution to the measurement of each one of the factors. For example, the item “Someone has threatened me” is one of the most representative ones of victimization. The item “I have said insulting words about someone to other people” is one of the most representative ones of aggression. The two-factor model suggests configural invariance depending on sex. Therefore, it can be considered as a sufficiently robust and generalizable model for girls and boys from Ecuador. These results align with the ones obtained in the original study of the scale in Spain [12] and in other European countries [11]. At the sight of the found evidence, the accuracy and quality of the measurement of the EBIPQ are confirmed in Ecuadorian adolescents.

The second objective was to establish the psychometric properties of the EBIPQ adaptation to measure the discriminatory bullying around SEN. The results referring to the IRT and CFA analyses show optimal psychometric properties for the initial structure of two factors—victimization and aggression—with accuracy and quality of the measurement. The quality of the measurement of each of the items and its contribution to the measurement of each one of the factors is also remarkable. For example, the item “Someone has threatened me” is one of the most representative ones of discriminatory victimization around SEN; the item “I have threatened someone” is one of the most representative ones of discriminatory aggression around SEN. Depending on gender, it has also been shown that the two-factor structure is generalizable. Bearing in mind so, we can confirm the hypothesis that the adaptation of the EBIPQ to measure the discriminatory bullying around SEN on the Ecuadorian student population shows an homogenous structure to the original instrument [11,12]. This way, we count with an instrument to which we will logically name European Bullying Intervention Project Questionnaire - Special Education Needs Discrimination Version (henceforth EBIPQ-SEND).

Regarding the third objective of the study, we conclude that around four out of every ten adolescents is involved in traditional bullying: two out of every ten as victims; one out of every ten as a bully; and two out of every ten as bully/victims. Both the prevalence of general involvement and the prevalence in every single role reported in Ecuador are homogenous with those reported in Latin American countries such as Nicaragua [14,15] and Colombia [13]. By contrasting the involvement by roles in Ecuador depending on gender, we can conclude that boys stand out for being bullies and girls for being bully/victims. Both conclusions are coherent with the findings in previous studies carried out in Nicaragua and Colombia [13,14]. We also conclude that girls stand out as victims. This is in agreement with the reported observations in Nicaragua [15] and in opposition to the ones reported in Colombia [13] in which boys stand out as victims. We conclude that the involvement in victimization in traditional bullying reported in Ecuador—victim and bully/victim—is slightly higher than the one in other countries of the Latin American area [13,14,15] and Spain [8]. The involvement in Ecuador is more than the double of the estimation for Europe and North America [16].

## 5. Conclusions

Regarding discriminatory bullying around SEN, we conclude that the same roles appear in discriminatory and traditional bullying. In discriminatory bullying around SEN, three out of every ten adolescents are involved: two out of every ten is a victim; one out of every ten is a bully; and between one or two out of every ten are bully/victims. Both in the role of discriminatory bully and discriminatory bully/victim around SEN, boys stood out more than girls. The observed pattern of predominance of boys in the discriminatory aggression around SEN is coherent with some observations carried out in previous studies (e.g., [29,30]). The findings in this study clearly show that intermediate adolescence, between 14 and 16 years old, is the stage of developmental development where there is greater harassment and intimidation around SEN, especially in the roles of victimization and victimized aggressor.

We observed a high correlation between the factors of victimization and aggression in traditional bullying. This suggests that there is a high or narrow co-occurrence of the roles of victim and bully. In fact, a significant percentage recognizes him/herself as a victim and as a bully. All of this is coherent with the findings in previous studies [11,12,13] and contributes to support the theory that there is certain dynamism in the roles within this phenomenon. For example, the adolescents who start being victims could become bully/victims over time. In the discriminatory bullying around SEN, we observed a high correlation between its factors of discriminatory victimization and discriminatory aggression. This is coherent with recent observations [21,22,23,28], so we may raise the possibility that there is also certain dynamism between roles in discriminatory bullying.

The validation of the EBIPQ to measure traditional bullying allows us to have an instrument for Ecuador and it can be added to the validations of the instrument in the Latin American countries (e.g., [13]), which have already shown certain strength and reliability in different European countries [11]. The use of this instrument, which is responsive to the integrated measurement of victimization and aggression, allows us to overcome the limitation of most of the instruments that has been responsive only to the report of victimization or aggression in a separate way [7]. Having a reliable instrument at our disposal that reports both dimensions of bullying is very important for the assessment and diagnosis of this violent phenomenon in which the dynamics of roles among peers is established.

The review of scientific literature has shown that the three research lines studying bullying in relation to SEN consider such violent phenomenon as something general: they study traditional bullying (e.g., [3,23,29,30]).By using the EBIPQ-SEND, we begin a new research line which specifically considers discriminatory bullying towards SEN. The fact that the EBIPQ-SEND is responsive to both discriminatory victimization and discriminatory aggression supposes a remarkable advantage regarding the original EBIPQ and other measurement instruments of traditional bullying in relation to school-aged children with disabilities or SEN. The EBIPQ-SEND will allow us to focus on the singularities of the aggression and victimization of this collective. This is a necessary condition to prevent and reduce the bullying in which they are involved [28,35]. Use of the EBIPQ-SEND may contribute to overcoming the limitations due to the scarcity of validated instruments to assess bullying in students with SEN [27], especially when is a discriminatory bullying [23]. The spread of the use of the EBIPQ-SEND could also contribute to overcoming the differences in the measurement criteria of the discriminatory bullying around SEN and to the cultural transferability in the operationalization of that construct [26]. Another advantage that this instrument provides is that, by being responsive to aggression and victimization, it will allow us to assess and diagnose discriminatory bullying around SEN as a dynamic violent phenomenon in which the role play among the peers involved is established.

The limitations of this study are associated with the use and application of self-reports and with the design of a transversal research. In further studies, it would be advisable to use hetero-reports as a measurement in addition to the self-reports. It would also be advisable to develop longitudinal studies by using validated questionnaires (EBIPQ and EBIPQ-SEND), which would allow us to improve our knowledge not only about traditional bullying but also about discriminatory bullying around SEN regarding the potential dynamism of roles, as well as the possible protection factors, predictors, and consequences. A strength of the study is that it has been developed over a wide sample that is representative of the population, a fact which is not common in the studies of these phenomena at a global level.

## Figures and Tables

**Figure 1 ijerph-16-00142-f001:**
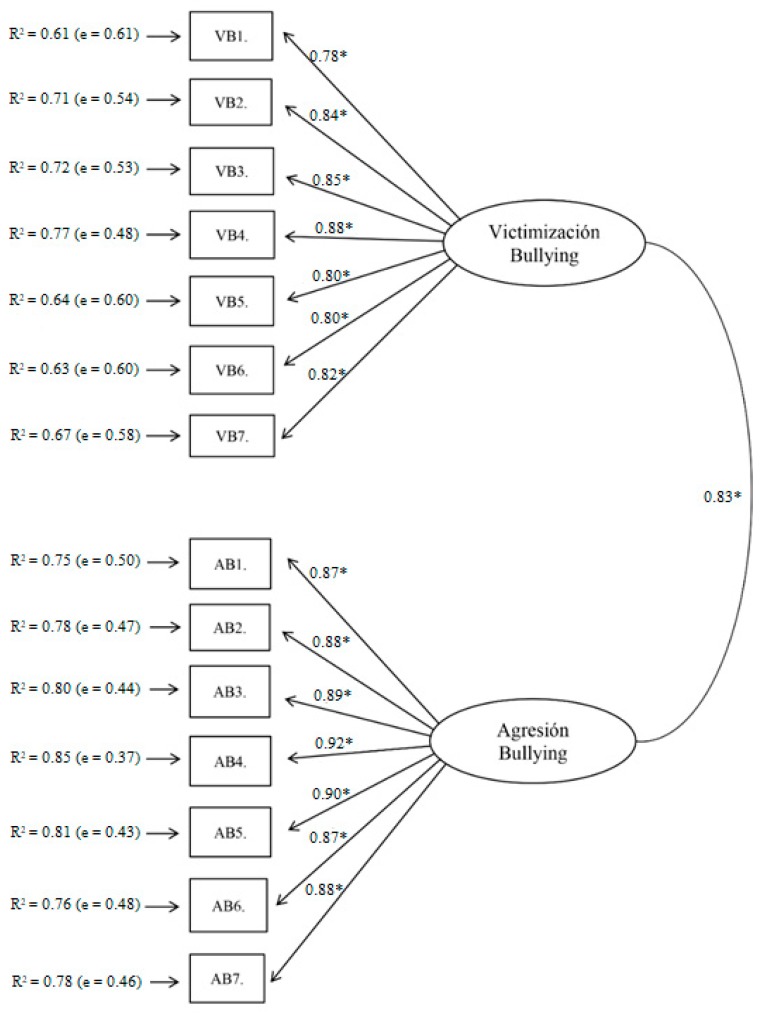
Confirmatory factorial analysis (CFA) of the adapted EBIPQ for Ecuador (* *p* < 0.05).

**Figure 2 ijerph-16-00142-f002:**
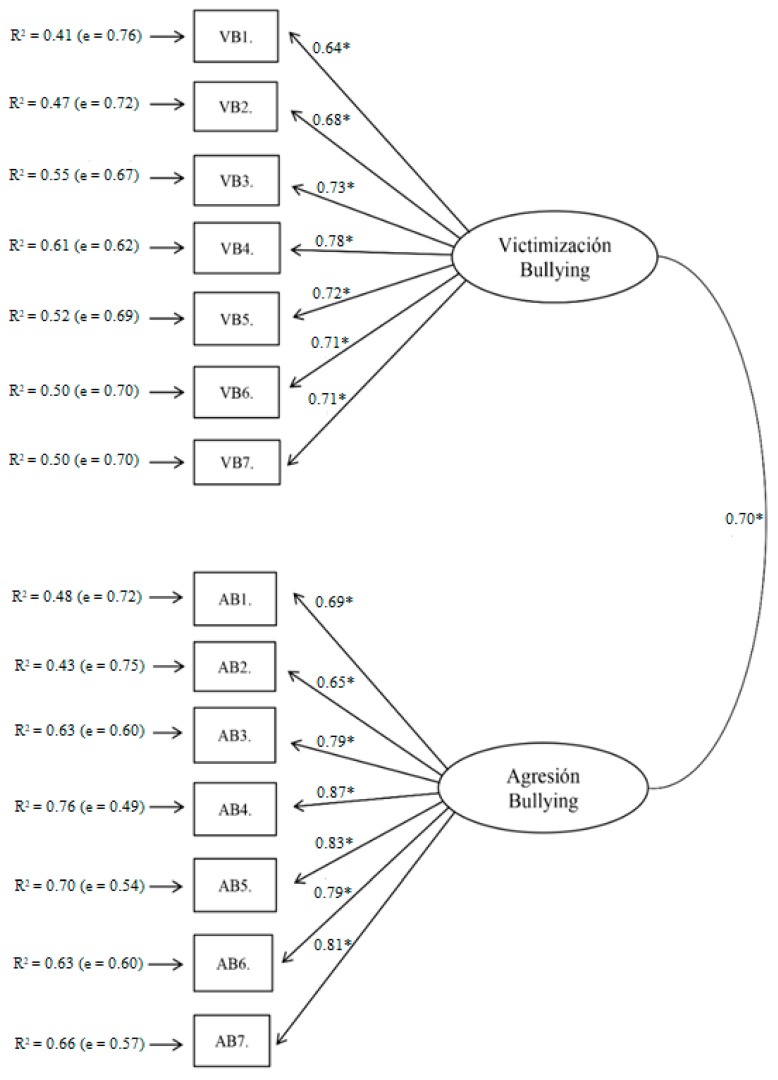
CFA of the adapted EBIPQ to report discrimination around special educational needs (SEN) (* *p* < 0.05).

**Table 1 ijerph-16-00142-t001:** Mean, standard deviation, skewness, kurtosis, and 4PL analysis (Item Response Theory, IRT).

Items (EBIPQ)		M	SD	Skew	Kurt	a	b	c
**VB1**	Someone has hit me, kicked me or pushed me	1.54	0.96	2.12	4.21	1.23	0.33	0.00
**VB2**	Someone has insulted me	2.04	1.24	1.31	0.75	1.97	−0.08	0.00
**VB3**	Someone has said insulting words about me to other people	1.89	1.13	1.40	1.26	0.81	−1.31	0.01
**VB4**	Someone has threatened me	1.48	0.96	2.28	4.67	1.34	1.78	0.00
**VB5**	Someone has stolen or broken my things	1.61	1.02	1.98	3.44	0.71	1.12	0.02
**VB6**	I have been excluded, isolated or ignored by other people	1.56	0.99	2.09	3.93	0.92	0.01	0.00
**VB7**	Someone has spread rumors about me	1.77	1.09	1.63	2.06	0.87	0.88	0.04
**AB1**	I have hit, hurt or pushed someone	1.62	1.00	1.93	3.35	1.89	0.45	0.00
**AB2**	I have insulted or said insulting words to someone	1.74	1.03	1.74	2.72	1.73	0.44	0.00
**AB3**	I have said insulting words about someone to other people	1.50	0.92	2.13	4.29	1.71	0.97	0.01
**AB4**	I have threatened someone	1.33	0.83	2.95	8.70	1.88	1.93	0.00
**AB5**	I have stolen or damaged something from someone	1.31	0.77	3.07	9.72	1.55	1.12	0.02
**AB6**	I have excluded, isolated or ignored someone	1.39	0.84	2.61	7.06	1.12	1.3	0.00
**AB7**	I have spread rumors about someone	1.34	0.84	2.79	7.90	1.65	1.14	0.02

Note: a = discrimination; b = difficulty; c = probability of failure (Random). EBIPQ = European Bullying Intervention Project Questionnaire; VB = Victimization Bullying; AB = Aggression Bullying; M = Mean; SD = Standard deviation.

**Table 2 ijerph-16-00142-t002:** Matrix of EBIPQ polychoric correlations.

Item	VB1	VB2	VB3	VB4	VB5	VB6	VB7	AB1	AB2	AB3	AB4	AB5	AB6	AB7
**VB1**	1													
**VB2**	0.55	1												
**VB3**	0.45	0.61	1											
**VB4**	0.49	0.49	0.57	1										
**VB5**	0.47	0.47	0.48	0.59	1									
**VB6**	0.43	0.44	0.50	0.56	0.53	1								
**VB7**	0.37	0.47	0.60	0.52	0.49	0.56	1							
**AB1**	0.53	0.46	0.35	0.46	0.44	0.38	0.39	1						
**AB2**	0.37	0.54	0.43	0.37	0.38	0.33	0.40	0.63	1					
**AB3**	0.41	0.40	0.45	0.50	0.43	0.42	0.46	0.60	0.64	1				
**AB4**	0.38	0.29	0.32	0.56	0.43	0.41	0.38	0.59	0.55	0.69	1			
**AB5**	0.39	0.27	0.30	0.51	0.47	0.42	0.38	0.53	0.44	0.62	0.77	1		
**AB6**	0.34	0.29	0.32	0.45	0.42	0.46	0.40	0.50	0.47	0.57	0.69	0.71	1	
**AB7**	0.37	0.32	0.35	0.47	0.41	0.42	0.45	0.49	0.48	0.63	0.71	0.71	0.71	1

Note: All the correlations with *p* < 0.01.

**Table 3 ijerph-16-00142-t003:** Gender configuration invariance analysis of the EBIPQ.

	Mod	χ^2S-B^	*Df*	*p*	NNFI	CFI	RMSEA	SRMR	∆χ^2S-B^	∆*p*	∆NNFI	∆CFI		∆RMSEA	∆SRMR
	General	1258.800	76	0.00	0.950	0.954	0.041	0.080							
**Gender**	Mod 1	1342.562	76	0.00	0.958	0.955	0.050	0.079	83.762	0.739 (n.s.)	0.008		0.001	0.009	0.001
	Mod 2	1292.941	76	0.00	0.960	0.961	0.051	0.077	34.141	0.831 (n.s.)	0.010	0.007		0.010	0.003

Note: Mod 1 = men; Mod 2 = women; n.s. = non-significant; CFI = comparative fit index; NNFI = non-normality fit index; RMSEA = root mean square error of approximation; SRMR = root mean square residual.

**Table 4 ijerph-16-00142-t004:** Role descriptive (Special Educational Needs -SEN-) by gender and age.

Role	Men %	Woman %	Early Adolescence % (11–13 Age)	Intermediate Adolescence % (14–16 Age)	Late Adolescence % (17–20 Age)
Victimization	52 (1.9)	48 (−1.9)	24.8 (−2.9)	60.4 *** (2.9)	14.8 (−2.9)
Aggression	54.6 ** (2.8)	45.4 (−2.8)	23.7 (−1.1)	58.5 (1.1)	17.8 (−1.1)
Victimized aggressor	57 *** (7.6)	43 (−7.6)	20.5 (−2.2)	58.6 ** (2.2)	20.9 (−2.2)

Note: ** *p* < 0.01; *** *p* < 0.001.

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
