# Peer review of "Traditional Bullying and Discriminatory Bullying Around Special Educational Needs: Psychometric Properties of Two Instruments to Measure It"

_ijerph, 2019, doi:10.3390/ijerph16010142_

Round 1

Reviewer 1 Report

The best parts of the paper are introduction and conclusions. The introduction provide sufficient background and include all relevant references. The conclusions supported by the results.

I propose a few changes:

- to add a date when the study was organized

- to add the name of the coordinating institution

- to write more about SEN measuring  - we do not know in what way the authors have asked students if they have/not to have SEN. If  they have not measured the SEN they should to add this information. And, in conclusion, write that their conclusions are rather general then resulting from the research.

- to add information how many students had had SEN. The authors have written" In discriminatory bullying around SEN, three out of every ten adolescents are involved (...)" but we do not know on what basis they have concluded that.

- to add information (I suggest to add a table) about bullying by gender and age. There is no information about a occurrence of bullying due to the age of the students.

- to add information about the web side of the project.

Author Response

Dear Reviewer 1,  We appreciate your kind cooperation in reviewing our manuscript and its valuable contributions.  The main changes are found in the description of the method - specifically in the instruments section - and in the presentation of the results - a table with the suggested data was added.  Some changes are also incorporated from the contributions of Reviewers 2 and 3. The change control was used in the new document to make the adjustments visible.  Attached is the document with the point-to-point answers of your recommendations.  Best regards,  The authors.

Reviewer 2 Report

The manuscript reports a study about traditional and discriminatory bullying in Special Educational  Needs.

Spell error: line 15 must be translated in to english "(Mage = 14.76, DT = 1.65)"

Introduction

Line 74 "with NEE" authors have not explained what NEE means, it is also an spell error?

Authors have presented a section about traditional bullying (1.1) but they don't present one fr discriminatory bullying. Moreover, both sections about bullying (1.1 and 1.2) should be summyrized. They are very rich on Information and literature Reviews but not all is relevant, as it is described.

Materials and methods

Sample size estimation was very well done, considering representativeness and it is well balanced regarding gender.

Instruments and procedures are well described.

Data Analysis

The statistical methods used very  strong and robust Tools, authors have presented all the fit indices that supports the final model.A plus on the Analysis: auhtors have choosen McDonald's Omega index to meausre internal consistency (instead of Chronbach's Alpha), very good!

Results

This section is very good written and results are presented in a very understandable way, graphics and statistics are didatic and easy to interpret.

I am very impressed by the methods choosen in the Analysis, they used more advanced and robust statistical procedures, congratulations on it.

Discussion

Here authors come again to theirs hypothesis and conclude them based on results, make some comparisson with literature and the final disclosure is achieved.

Author Response

Dear Reviewer 2,

We appreciate your kind cooperation in reviewing our manuscript and its valuable contributions.

The main changes are found in the description of the method - specifically in the intruments section - and in the presentation of the results - a table was added- . Some changes are also incorporated from the contributions of Reviewers 1 and 3.

The change control was used in the new document to make the adjustments visible.

Attached is the document with the point-to-point answers of your recommendations.

Best regards,

The authors.
